# Comparison of Wind Tunnel Test Data for Low-Rise Buildings with Main Wind Force Resisting System Design Procedures

**S. M. Ashfaqul Hoq** [1] **and Johnn P. Judd** [2,*]

1   Department of Civil and Environmental Engineering, North South University, Plot 15, Block B, Bashundhara R/A, Dhaka 1229, Bangladesh; ashfaqul.hoq@northsouth.edu
2   Department of Civil and Construction Engineering, Brigham Young University, 430 EB, Provo, UT 84602, USA
*   Correspondence: johnn@byu.edu

**Abstract:** The adequacy of the directional and envelope procedures for the design of the main wind force resisting system is not well understood. The purpose of this study is to evaluate the directional and envelope procedures based on wind tunnel test data for a set of low-rise enclosed buildings with gable-shaped roofs in open terrain (Exposure C). The base shear force and the conditional reliability index are used to determine the adequacy of the procedures. The base shear was compared to the design base shear in each direction based on the horizontal component of the wind load on the wall and roof. The reliability index, $\beta$ conditional on the occurrence of the design wind speed was computed for a range of system capacities. The main findings are (1) the directional procedure produced a larger design base shear compared to the envelope procedure, primarily due to the difference in external pressure coefficients, (2) the directional procedure provided a higher $\beta$, and (3) the envelope procedure provided a $\beta$ that did not meet the standard target $\beta$ equal to 3.0 for the main wind force resisting systems with low variability in capacity, but neither procedure met the standard target $\beta$ for the main wind force resisting systems with high variability in capacity.

**Keywords:** wind loads; main wind force resisting system; low-rise buildings

## 1. Introduction

The design of the main wind force resisting system (MWFRS) in buildings is a basic part of structural engineering practice, but currently there is no consensus on the appropriate procedure to determine MWFRS design wind loads for low-rise buildings. For instance, there are three procedures (the "directional", "envelope", and "wind tunnel" procedures) in ASCE 7–16, Minimum Design Loads and Associated Criteria for Buildings and Other Structures [1]. In the directional and envelope procedures, design loads are determined analytically using pressure or force coefficients that are primarily based on data from wind tunnel tests of generic building models. In the wind tunnel procedure, design loads are determined experimentally in a wind tunnel using pressures measured on the surface of a small-scale model of the actual building that is being designed. Thus, the wind tunnel procedure is primarily used for flexible structures, irregularly shaped buildings, buildings shielded by adjacent buildings, or other structures that require special considerations, whereas the directional and envelope procedures are used for routine design. Accordingly, this study focuses on the directional and envelope procedures.

The directional and envelope procedures are fundamentally different. In the directional procedure (ASCE 7–16, Chapter 27), design loads are based on external pressure coefficients that are applied normal to the wall or roof surface for a specific wind direction. Hence, it is designated the "directional" procedure. In the envelope procedure (ASCE 7–16, Chapter 28), design loads are based on pseudo-external pressure coefficients that are applied to a vertical or horizontal projection of the wall or roof surface. The pseudo-external pressure coefficients were derived by rotating the building model 360 degrees in the wind tunnel while simultaneously monitoring the wall and roof pressures. The

magnitude of the external pressure coefficients that envelope the maximum structural response (uplift forces, base shear, and bending moments in building frames) are recorded and used to determine the pseudo-external pressure coefficients [2]. Consequently, the pressure coefficients in the envelope method do not correspond to any particular wind direction. Instead, the pressure coefficients envelope the structural response. Hence, it is designated the "envelope" procedure. As might be expected, the differences between the two procedures leads to different design loads.

The difference in design loads produced by the directional and envelope procedures depends primarily on the geometry of the building (e.g., [3,4]). The envelope procedure produces lower design wind pressures in general [1], and these lead to lower design loads, compared to the directional procedure. To illustrate, consider two typical building types shown in Table 1. The first type is a residential building. The building was designed by a national home builder and actually constructed. The building is representative of residential structures in the United States built during the past two decades. The second type is an office building. The building is a well-known example building used for seismic design provisions [5]. The story forces produced by the directional and envelope procedures for these two building types are compared in Table 1. For the residential building, the directional procedure story forces are 11% larger than the envelope procedure story forces. For the office building, the difference is even greater. The directional procedure story forces are 50% larger, on average. Similar substantial differences in design loads have been reported for other structures in the literature (e.g., [6]).

Prior research has focused primarily on the spatial distribution of wind loads produced by the directional and envelope procedures compared to wind tunnel data. He et al. [7] assessed the adequacy of the directional and envelope procedures in ASCE 7–10 [8] for low-rise wood frame structures based on wind tunnel test data. The study showed that the envelope procedure produces design uplift loads that more closely match the forces in roof sheathing-to-framing connections in the wind tunnel tests, but the directional procedure produces design loads that more closely match the spatial distribution of load in the wind tunnel tests.

**Table 1.** Ratio of story force produced by the directional procedure to the envelope procedure.

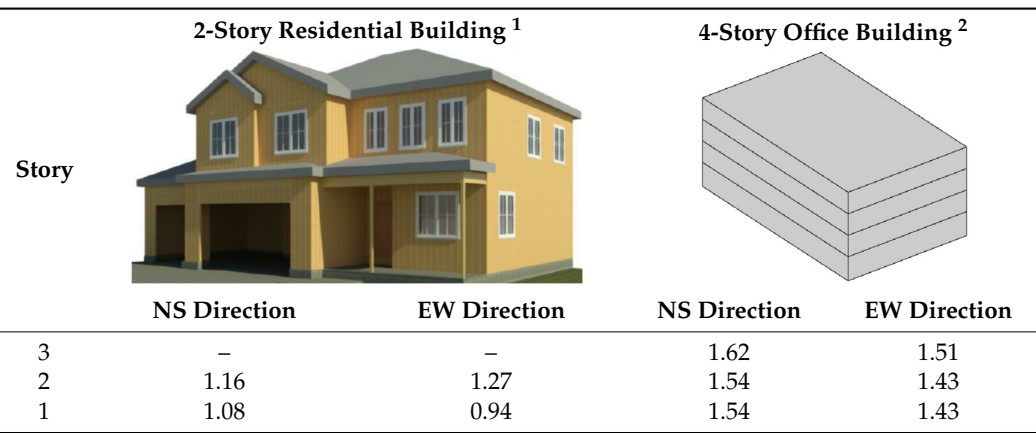

| Story | 2-Story Residential Building [1] | | 4-Story Office Building [2] | |
|---|---|---|---|---|
| | NS Direction | EW Direction | NS Direction | EW Direction |
| 3 | – | – | 1.62 | 1.51 |
| 2 | 1.16 | 1.27 | 1.54 | 1.43 |
| 1 | 1.08 | 0.94 | 1.54 | 1.43 |

[1] The first story is 16.5 m wide by 12.8 m deep (including a third-car garage). The second story is 12.2 m wide by 12.5 m deep. The roof is a hip shaped over the main area and gable-shaped over the front. The mean roof height is equal to 7.92 m. [2] The building is 38.1 m wide by 24.4 m deep by 12.2 m tall, with a 4.57 m first story height, and 3.66 m upper story height. The roof is flat.

Nevertheless, analysis of the wind tunnel test data [9] included in the NIST aerodynamic database [10,11] shows that the envelope procedure actually underestimates both wind loads [12] and wind load effects [13,14]. The underestimated loads are attributed to the lower turbulence intensity and pressure tap density in the more recent wind tunnel tests, compared to the tap density in the legacy wind tunnel tests, which were used in the development of the envelope procedure [15,16]. Recently, Wang and Kopp [17] compared the directional and envelope procedures to wind tunnel test data and concluded, among

other findings, that the envelope procedure tends to underestimate the windward wall pressures for buildings with a larger height-to-length ratio. The effect of MWFRS capacity on the reliability of the procedures was not examined.

Differences in design loads are significant, not only because they can lead to drastically different structural designs, but because the differences undermine confidence in the wind provisions themselves. For instance, a lack of confidence in the significant change to ASCE 7–16 wind provisions for components and cladding of low-slope roofs [18,19] was a likely contributor to the controversy regarding adoption of ASCE 7–16 into the building code (e.g., [20,21]). For these two reasons, it is imperative to compare the MWFRS design procedures with experimental data, and to determine their adequacy.

This study evaluates the directional and envelope procedures based on wind tunnel test data of low-rise buildings with gable-shaped roofs in open terrain (Exposure C). The hypothesis is that the nominal MWFRS capacity can be equated to the strength-level design MWFRS loads, and therefore the reliability index, $\beta$ conditional on the occurrence of the design wind speed, can be used to evaluate the procedures and account for variability in MWFRS capacity. To test the hypothesis, the horizontal component of the wind load on each surface of the building was computed, and the resultant base shear forces were determined for the set of buildings described in the next section.

## 2. Approach

### 2.1. Building Description

Figure 1 shows the types of buildings examined in this study. The buildings were 38.1 m long by 24.4 m wide, enclosed, and had a gable-shape roof with a 26.6° roof slope (6:12 roof pitch). Two eave heights were considered: 3.66 m and 12.2 m. The building dimensions correspond to small-scale building models that were tested in the atmospheric boundary layer wind tunnel at Western University in London, Ontario, Canada [11]. The wind tunnel model was instrumented with pressure taps. Figure 2 shows the pressure tap layout. There were 485 pressure taps for the 3.7-m (12-ft) eave height building, and 701 pressure taps for the 12.2-m (40-ft) eave height building. The wind pressure was measured at each tap and the corresponding external pressure coefficient, $C_p$, was defined as the wind pressure (minus the static pressure) divided by the velocity pressure measured at the region of uniform flow in the wind tunnel above the model ("upper level" dynamic pressure in the wind tunnel). The wind direction varied from parallel to ridge (0°) to normal to ridge (90°). The sampling rate was 500 Hz. The duration was 100 s in the tests. The equivalent sampling rate for the full-scale building is 17 Hz, and the equivalent full-scale duration is 0.8 h.

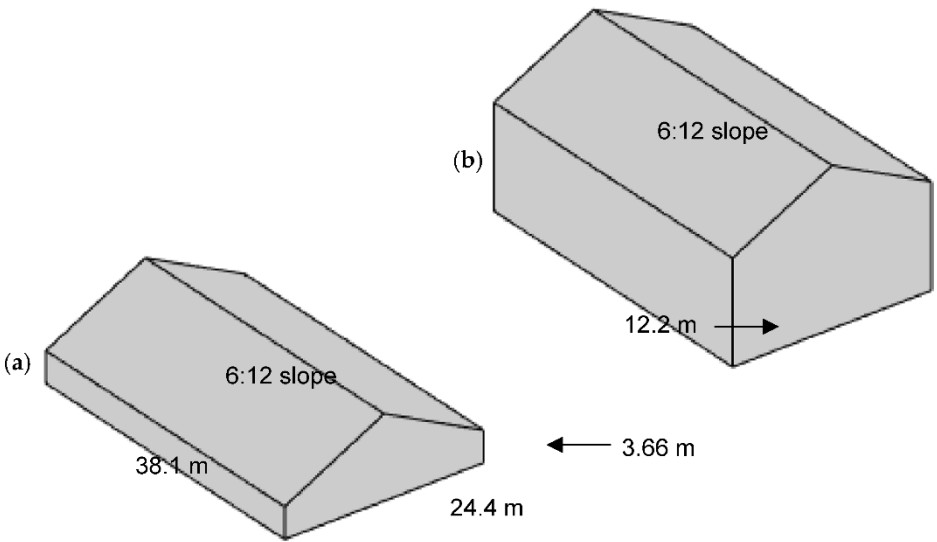

**Figure 1.** Buildings examined: (**a**) 3.66 m eave height; (**b**) 12.2 m eave height.

Three wind directions were examined: wind parallel to ridge, oblique wind (wind at 45° to the ridge), and wind normal to ridge. Wind parallel and normal to ridge were selected because these are the principal directions of the building. The angle of the oblique wind was based on prior research that shows 40° to 50° is critical for typical low-rise buildings with gable-shape or hip-shape roofs [22].

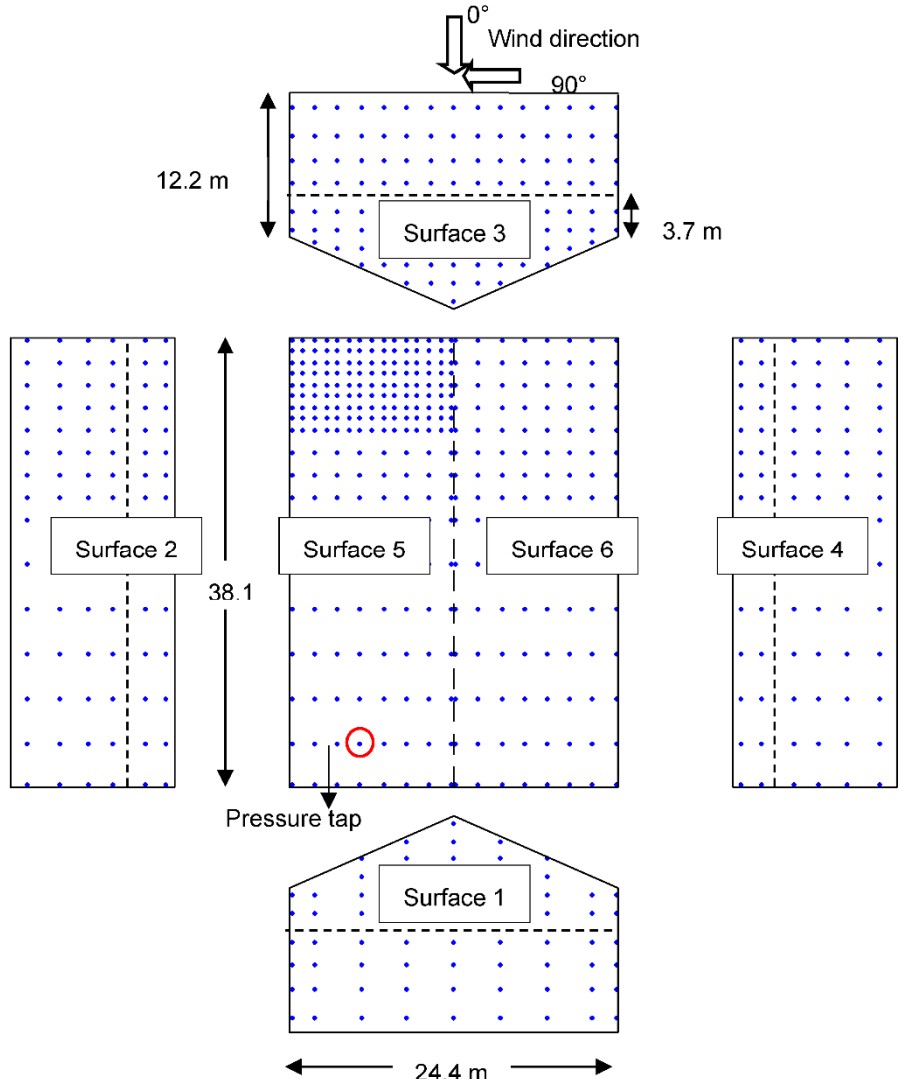

**Figure 2.** Pressure tap layout in the wind tunnel model.

Figure 3 shows the spatial distribution of $C_p$ from the wind tunnel tests for wind parallel to ridge for the 12.2-m eave height building. For comparison with ASCE 7–16, the $C_p$ values shown in Figure 3 are "equivalent" ASCE 7–16 values of $C_p$. An equivalent value of $C_p$ was calculated by removing the effects of terrain exposure and building height from the wind tunnel data [12], and by using the averaging time in ASCE 7–16 (a 3-s gust wind speed) instead the average time used in the wind tunnel tests (an hourly wind speed). The resulting conversion factor from the test data $C_p$ to an equivalent ASCE 7–16 $C_p$ was 1.08 for the 3.7-m eave height building, and 0.91 for the 12.2-m eave height building. On the roof, $C_p$ varies from approximately −0.40 near the windward edge, to −0.05 on the leeward area. On the sidewalls, $C_p$ varies from approximately −0.35 near the windward edge to 0 on the leeward area. On the windward wall, $C_p$ varies from approximately 0.40 at the center to approximately 0.15 at the edges. The corresponding value of $C_p$ in ASCE 7–16 was multiplied by the gust-effect factor, $G$ (0.85).

In general, the directional and envelope procedures produced larger values of $C_p$ compared to the mean value from the wind tunnel tests. However, for the buildings examined in this study, the envelope procedure better captured the spatial distribution of wind pressure because the envelope procedure includes a critical edge zone on the windward surfaces.

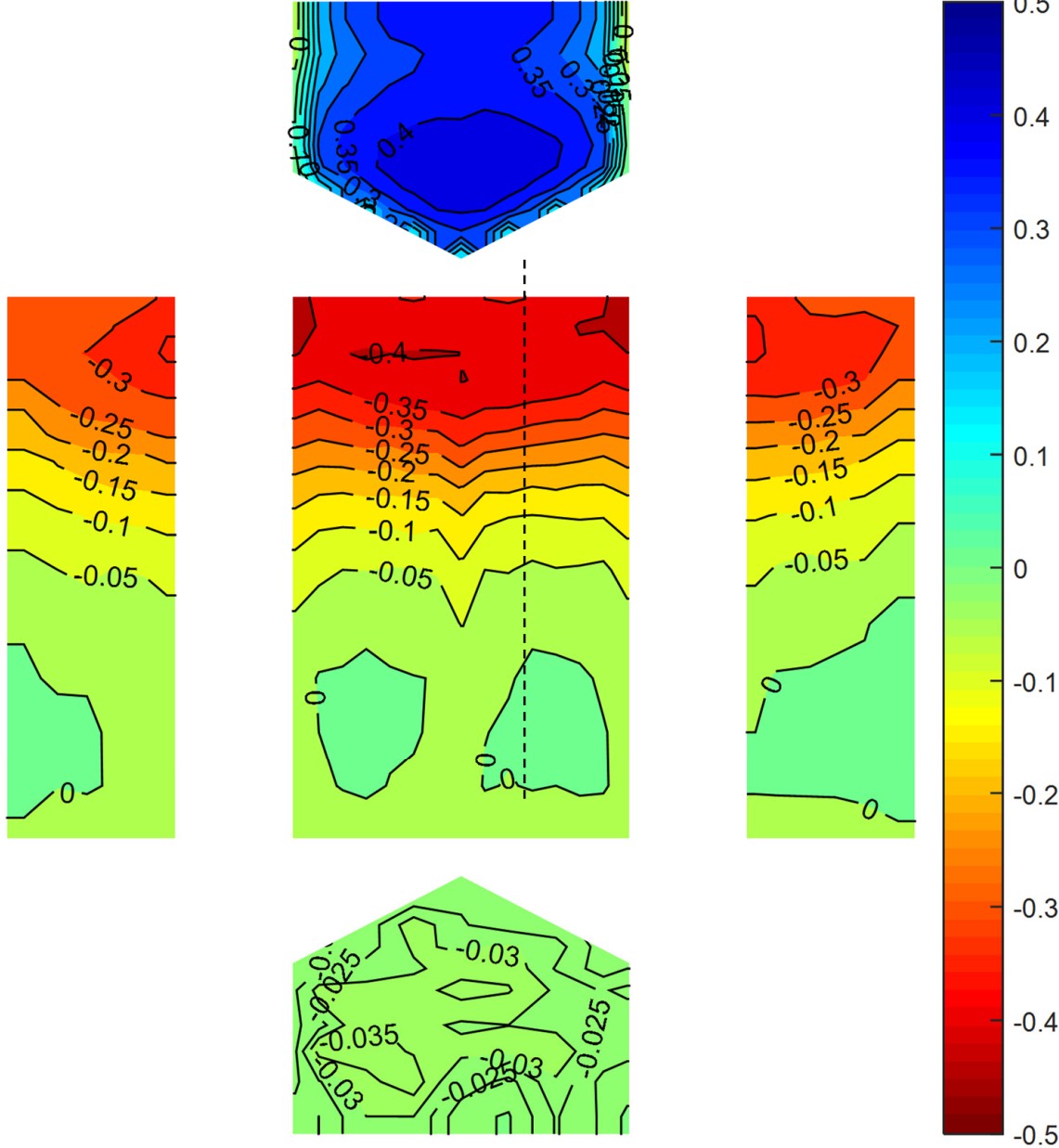

**Figure 3.** Spatial distribution of mean $C_p$ from the wind tunnel tests for wind parallel to ridge for the 12.2-m eave height building.

Using the directional procedure, $C_p$ is $-0.9$ on near the windward edge, $-0.7$ on the side wall and $0.8$ on the windward wall (ASCE 7–16 Figure 27.3–1). Using the envelope procedure, near the edge, $GC_p$ is equal to $-1.07$ on the windward roof, $-0.53$ on the leeward roof, $-0.48$ on the sidewall containing the critical edge zone, $-0.45$ on the side wall not containing the critical edge zone, $0.61$ on the windward wall containing the critical edge zone, and $0.4$ on the windward wall not containing the critical edge zone (ASCE 7–16 Figure 28.3–1).

### 2.2. Design Base Shear

The design loads for the MWFRS were calculated for three directions: normal to ridge (90°), parallel to ridge (0°), and wind at a 45° direction. Following ASCE 7–16, the velocity pressure, $q_z$ at height $z$ above the ground was calculated using Equation (1):

$$q_z = 0.613 K_z K_{zt} K_d K_e V^2 \tag{1}$$

where $K_z$ is the velocity pressure exposure coefficient, $K_{zt}$ is the topographic factor (1.0), $K_d$ is the wind directionality factor (0.85), $K_e$ is the ground elevation factor (1.0), and $V$ is the basic wind speed. In this study, $V$ was taken equal to 47.8 m/s (107 mph) because that corresponds to a typical basic wind speed in the interior of the United States for Risk Category II buildings.

The design wind pressure, $p_{design}$ was calculated using Equation (2):

$$p_{design} = q_z \left[ (GC_p) - (GC_{pi}) \right], \tag{2}$$

where $G$ is the gust-effect factor (0.85), $C_p$ is the external pressure coefficient, and $GC_{pi}$ is the combined internal pressure coefficient and gust-effect factor. In the directional procedure, $C_p$ was determined using ASCE 7–16, Figure 27.3–1 for wind parallel and normal to the ridge. In the envelope procedure, $GC_p$ was determined using ASCE 7–16, Figure 28.3–1. The envelope procedure, which is intended to envelope the structural response, involves two load cases: wind normal to ridge to wind 45° to ridge (Load Case A), and wind varying from parallel to ridge to 45° to ridge (Load Case B).

The design base shear was calculated based on the horizontal thrust forces contributed by the windward and leeward surfaces. Figure 4 shows the convention to define the base shear force along the left and right walls of the building. In the directional procedure, $C_p$ values are provided for only two directions: wind parallel and normal to ridge. Therefore, for wind at a 45° direction, the design base shear was taken equal to 75% of design bases shear for wind parallel to ridge plus 75% of design base shear for wind normal to ridge, similar to the approach used in prior research (e.g., [23]).

The wall height tributary to the roof diaphragm was taken as half of the wall height. For wind parallel to ridge, the windward wall (Surface 3) and the leeward wall (Surface 1) contribute to the base shear. For wind normal to ridge, the windward wall (surface 4), the windward roof (Surface 6), the leeward wall (Surface 2), and the leeward roof (Surface 5) contribute to the base shear. Since the buildings were enclosed, the horizontal thrust exerted by internal wall pressures cancels out and does not affect the base shear.

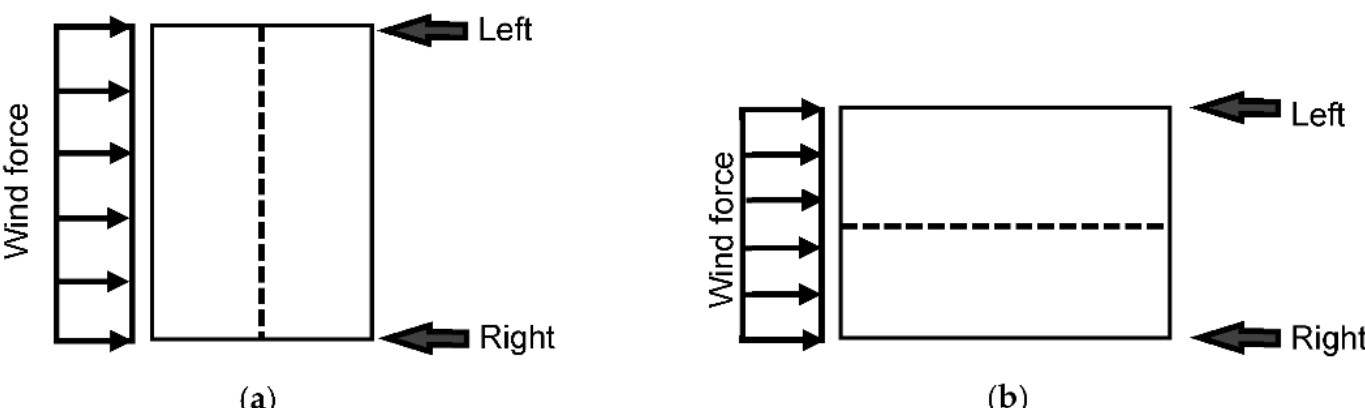

**Figure 4.** Convention to define the base shear force: (**a**) wind normal to ridge, and (**b**) wind parallel to ridge.

### 2.3. Base Shear Based on Wind Tunnel Test Data

The wind pressure based on wind tunnel test data, $p_{wt}$ was calculated using Equation (3):

$$p_{wt} = 0.5\rho V_{t,h,z_o}^2 C_p \tag{3}$$

where, $\rho$ is the air density (1.225 kg/m$^3$), $V_{t,h,z_o}$ is the mean wind speed (averaging over time $t$, at a reference height $h$, with a roughness length $z_o$), and $C_p$ is the pressure coefficient measured in the wind tunnel test.

In the wind tunnel test data, $V_{t,h,z_o}$ is a mean hourly wind speed value, and $C_p$ values are referenced to the mean hourly wind speed at the upper level of the wind tunnel. Therefore, to determine the base shear, the basic wind speed corresponding to a 3-s gust at 10-m was converted to mean hourly wind speed using a statistical correlation between wind speed and averaging time, commonly known as the "Durst curve" (ASCE 7–16, Figure C26.5–1). The value of $C_p$ was adjusted to match the 10-m reference height used in ASCE 7–16 using the velocity profile of the wind tunnel test.

The tributary area for each pressure tap was calculated based on the location of adjacent pressure taps and the distance to the edge of the surface. For example, Figure 5 shows the tributary areas for the 12.2-m eave height building. The load contributing from the pressure taps were summed to obtain the horizontal thrust contributed by the building surface at each step in the load history. The base shear was calculated as the sum of the contributions to horizontal thrust from windward and leeward surfaces. Drag forces on the surfaces parallel to the wind were small and neglected in the base shear calculation.

### 2.4. Reliability Index

The reliability index, $\beta$ conditioned on the occurrence of the design wind speed, was calculated using Equation (4) [24,25]:

$$\beta = \frac{\overline{R} - \overline{Q}}{\sqrt{\sigma_R^2 + \sigma_Q^2}} \tag{4}$$

where, $\overline{R}$ is the mean MWFRS capacity, $\sigma_R$ is the corresponding standard deviation in capacity, $\overline{Q}$ is the mean demand (calculated wind load based on the wind tunnel tests data), and $\sigma_Q$ is the corresponding standard deviation in demand. The nominal MWFRS capacity, $R_n$ was equated to the strength-level design MWFRS loads based on a demand-to-capacity ratio equal to 1.0. In the directional procedure, MWFRS loads for the 45° wind direction are equal to 75% of wind loads for wind parallel to the ridge, plus 75% of the wind loads for wind normal to the ridge. However, since the maximum design wind load in a given direction governs design, the design loads for the 45° wind direction do not control it. Consequently, the values $R_n$ and $\beta$ were calculated based on wind normal to ridge and wind parallel to ridge.

Variability in MWFRS capacity was included in the analysis using a range of values for the mean-resistance to nominal-resistance ratio, $\overline{R}/R_n$ and the coefficient of variation (COV) in capacity. In this study, two pairs of $\overline{R}/R_n$ and COV were considered: $\overline{R}/R_n$ equal to 1.1 and a COV equal to 0.1, for a MWFRS that exhibits a relatively low degree of variability, and $\overline{R}/R_n$ equal to 2.0 and a COV equal to 0.3 for a MWFRS that exhibits a relatively high degree of variability in capacity. The values for low variation in capacity are typical of values generally reported in the literature for structural systems utilizing materials with low variability in design properties (e.g., [26,27]). Similarly, the values for high variation in capacity are typical of values have been reported in the literature [28–30], particularly for light-frame wood shear walls [31–34]. Nevertheless, the pairs of $\overline{R}/R_n$ and COV used in this study were solely intended to represent a realistic range in MWFRS capacity.

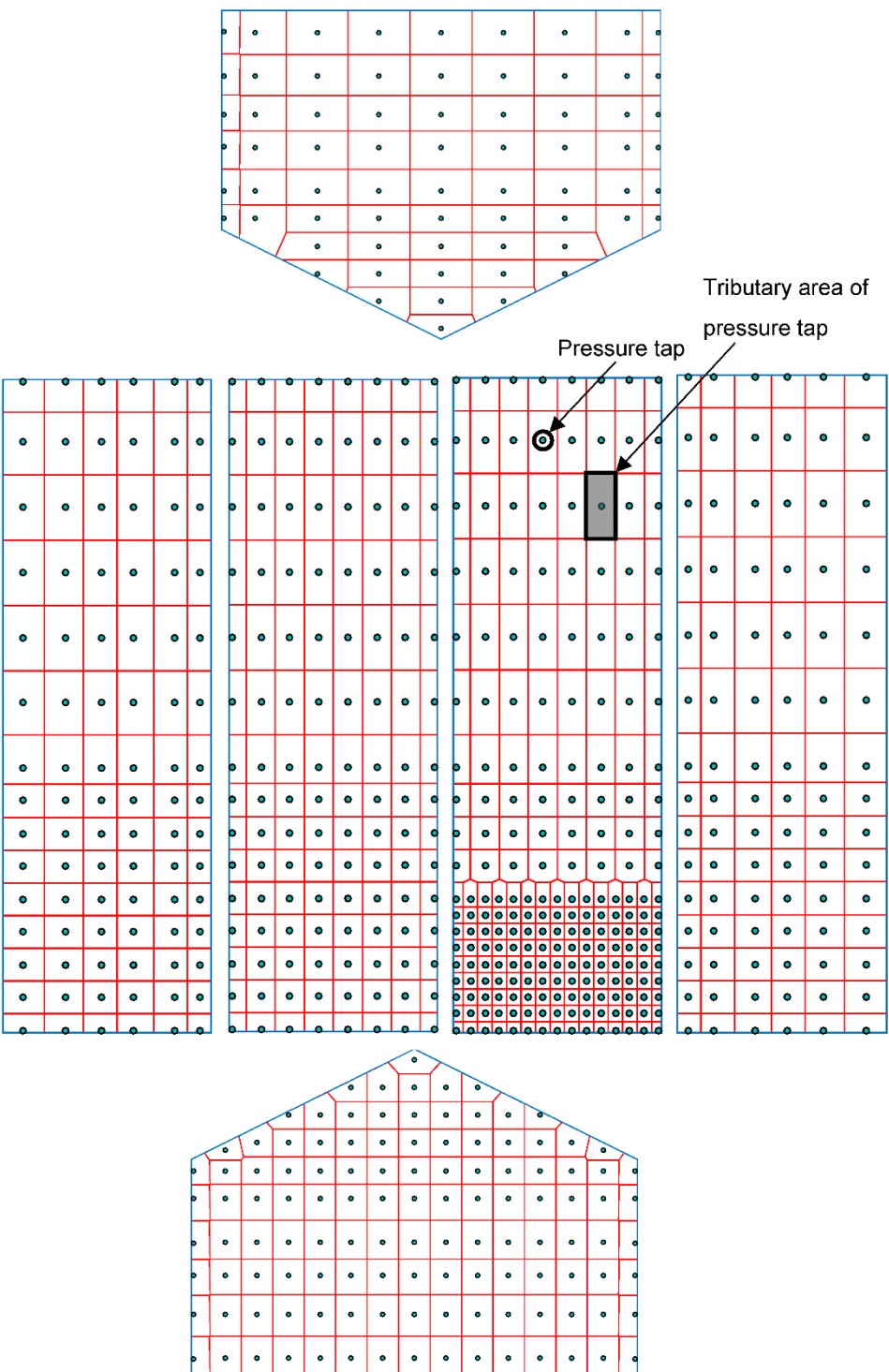

**Figure 5.** Pressure tap tributary area for 12.2-m eave height building.

## 3. Results

### 3.1. Base Shear

The design base shear is summarized in Table 2. The left reaction, right reaction, and total reaction (base shear) are given for the three wind directions and the two MWFRS orientations (wind parallel and perpendicular to ridge). For purposes of comparison, the horizontal thrust was resolved into left and right reactions, assuming that the vertical elements of the MWFRS are located on the perimeter of the building. For the directional

procedure, the left reaction is equal to the right reaction. For the envelope procedure, the reactions are the maximum values from the eight loading patterns shown ASCE 7 Figure 28.3–1 (i.e., both Load Cases A and B were applied). The larger reaction corresponds to the windward corner of the building containing the "edge strip" defined in ASCE 7–16. The base shear for oblique wind using the envelope procedure does not apply because the larger of the base shears for 0° and 90° would be used in design.

The design base shear produced by the directional procedure was larger than the base shear produced by the envelope procedure, as expected (e.g., Table 1). For wind normal to ridge and the MWFRS oriented parallel to wind, the base shear produced by the directional procedure was 64% larger compared to the base shear produced by the envelope procedure. The difference in base shear reflects the difference in external pressure coefficients. For example, in the directional procedure for wind normal to ridge, and for a small height-to-length ratio and a roof slope greater than 15°, there are two loading cases for the windward roof: positive pressure or negative pressure (suction). In contrast, the envelope procedure has only one load case for the windward roof: suction. This incongruity in the MWFRS procedures on predicting the direction of pressure (toward or away from the roof surface) is due to the transition from negative external pressure coefficient values for intermediate roof slope (a roof slope equal to 20°) to positive values for comparatively steeper slope roof (a roof slope ranging from 30° to 45°). Based on linear interpolation, the boundary between these roof slopes is equal to 28°, when the pressure changes from negative to positive values. The buildings considered in this study have a roof slope equal to 26.6°. Thus, it follows that in the envelope procedure, it is assumed that negative pressures are developed on the windward roof. Accordingly, the negative pressures on the windward roof lower the total horizontal thrust for wind normal to ridge. Furthermore, the contribution of the windward roof to base shear is greater for the buildings with a lower eave height. Another cause for difference between the procedures is that in the directional procedure, individual values of $C_p$ and $G$ are determined, whereas in the envelope procedure, a single combined value of $GC_p$ is used.

**Table 2.** Design base shear (kN).

| Wind Direction | MWFRS Orientation | Reaction Force | MWFRS Procedure | | | |
| --- | --- | --- | --- | --- | --- | --- |
| | | | Directional | | Envelope | |
| | | | Eave Height | | | |
| | | | 3.7 m | 12.2 m | 3.7 m | 12.2 m |
| Normal to Ridge (90°) | Normal to Ridge | Total | 273 | 513 | 166 | 405 |
| | | Left | 137 | 257 | 80.6 | 195 |
| | | Right [1] | 137 | 257 | 85.0 | 209 |
| Oblique (45°) | Normal to Ridge | Total | 205 | 385 | – | – |
| | | Left | 103 | 192 | – | – |
| | | Right [1] | 103 | 192 | – | – |
| | Parallel to Ridge | Total | 96.1 | 211 | – | – |
| | | Left [1] | 48.1 | 105 | – | – |
| | | Right | 48.1 | 105 | – | – |
| Parallel to Ridge (0°) | Parallel to Ridge | Total | 128 | 281 | 92.5 | 208 |
| | | Left [1] | 64.1 | 141 | 47.3 | 108 |
| | | Right | 64.1 | 141 | 45.0 | 100 |

[1] For the envelope procedure, the reaction force contains the contribution from the "edge strip" defined in ASCE 7–16.

The calculated base shear from wind tunnel test data is summarized in Table 3. The mean, $\mu$ and the standard deviation, $\sigma$ of the reactions and base shear are given for the three

wind directions and the two MWFRS orientations. Previous studies (e.g., [13]) tabulated both observed and estimated peak response. Since a consensus on the proper approach to estimate the statistical peak has not yet been established [35,36], this study instead uses the mean and the standard deviation of the wind load history, as described in the following section.

Notably, the base shear was larger for the 45° wind direction compared to wind normal ridge, for the MWFRS oriented normal to the ridge. The larger base shear was caused by the higher suction on the leeward roof. Similar behavior has been reported for gable roof buildings with a moderate slope (roof slope less than 30°) [22]. For the 90° and 45° wind directions, positive pressures were developed on the windward roof.

Comparison of the design base shear (Table 2) with the base shear based on wind tunnel test data (Table 3) shows that the envelope procedure underestimated the base shear by 22%, on average. In some instances, the directional procedure also underestimated the base shear (for example, in the 45° wind direction). For example, Figure 6 shows the design wind load history for the 3.7 m eave height building for wind normal to ridge. The envelope procedure produced larger design wind pressures on the windward corners of the building that contain the critical edge zone. The side of the building that contains the windward corner had higher reaction forces compared to the other side of the building.

**Table 3.** Base shear (kN) from wind tunnel test data.

| Wind Direction | MWFRS Orientation | Reaction Force | Eave Height | | | |
|---|---|---|---|---|---|---|
| | | | 3.7 m | | 12.2 m | |
| | | | $\mu$ | $\sigma$ | $\mu$ | $\sigma$ |
| Normal to Ridge (90°) | Normal to Ridge | Total | 104 | 25.8 | 212 | 56.4 |
| | | Left | 51.9 | 14.4 | 106 | 30.6 |
| | | Right | 52.7 | 14.6 | 108 | 31.1 |
| | Parallel to Ridge | | | | | |
| Oblique (45°) | Normal to Ridge | Total | 100 | 30.5 | 219 | 59.3 |
| | | Left | 36.2 | 12.9 | 85.8 | 25.7 |
| | | Right | 64.4 | 19.4 | 135 | 36.5 |
| | Parallel to Ridge | Total | 37.2 | 11.1 | 98.4 | 30.2 |
| | | Left | 24.8 | 7.50 | 61.9 | 19.3 |
| | | Right | 12.5 | 4.60 | 36.4 | 12.4 |
| Parallel to Ridge (0°) | Normal to Ridge | | | | | |
| | Parallel to Ridge | Total | 47.6 | 15.1 | 125 | 36.1 |
| | | Left | 23.9 | 8.30 | 62.7 | 19.7 |
| | | Right | 23.7 | 8.30 | 61.9 | 19.3 |

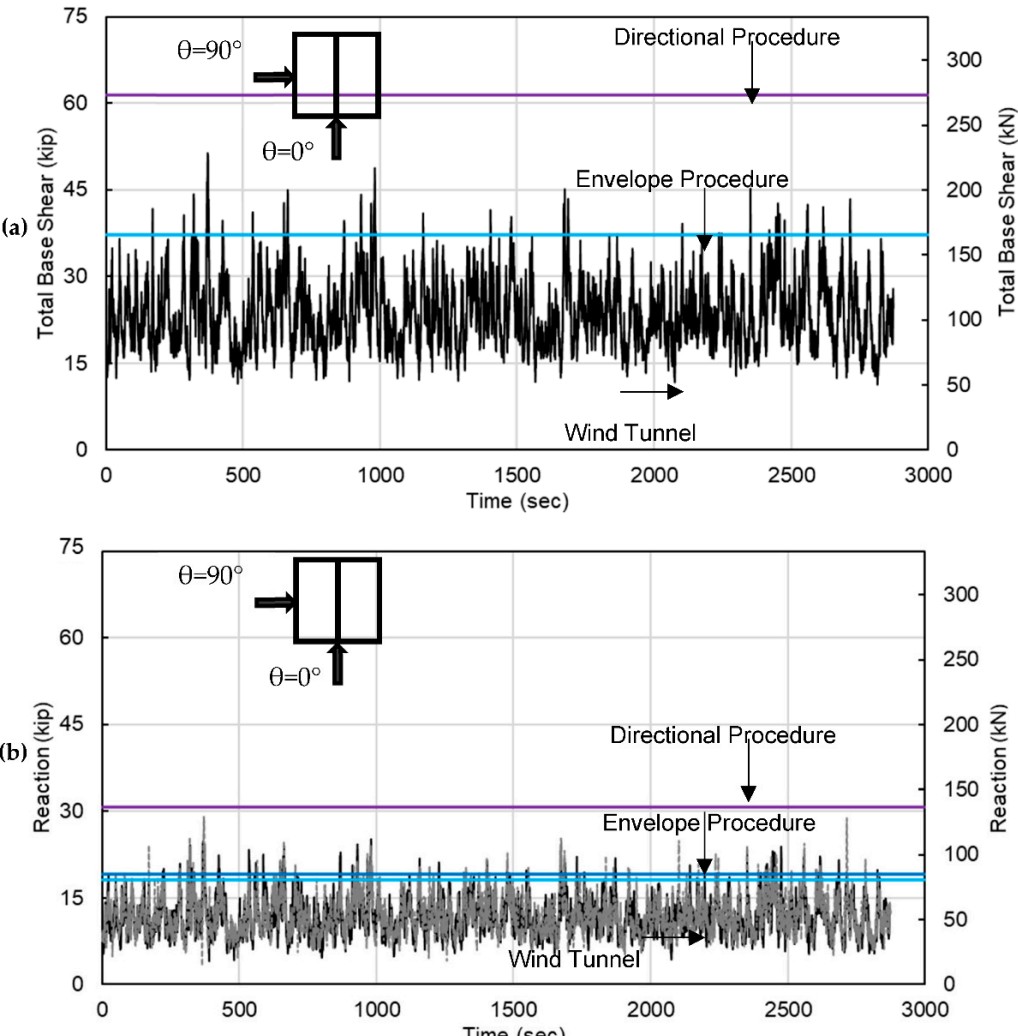

**Figure 6.** Load history for wind normal to ridge ($\theta$ = 90°) for 3.7 m eave height building: (**a**) base shear, and (**b**) reaction force.

### 3.2. Reliability Index

The values of $\beta$ are summarized in Table 4. Three values of $\beta$ were calculated: one for the resultant base shear, one for the left reaction, and one for the right reaction. A target $\beta$ was selected based on Risk Category II buildings (ASCE 7–16 Table 1.3–1). In ASCE 7–16, the target annualized probability of failure, $P_f$ is equal to $3.0 \times 10^{-5}$/yr. This corresponds to a target $\beta$ equal to 3.0, which is similar to target values of $\beta$ reported in the literature [24,25]. For the envelope procedure, the maximum base shear from Load Cases A and B are compared to the based shear based on the wind tunnel data in the 45° wind direction. The controlling value of $\beta$ corresponding to the critical cases for each procedure is underlined in Table 4.

The results indicate that the directional procedure provides a higher $\beta$ compared to the envelope procedure. The results also indicate that the minimum $\beta$ for the directional procedure (3.20) meets the target reliability for a MWFRS with low variability in capacity, but the minimum $\beta$ for the envelope procedure (1.35) does not. For a MWFRS with high variability in capacity, neither procedure has a minimum $\beta$ that meets the target reliability (2.39 for the directional procedure, and 1.93 for the envelope procedure).

**Table 4.** Reliability index, $\beta$ for MWFRS with a range of variability in capacity.

| Wind Direction | MWFRS Orientation | Base Shear | Directional Eave Height | | Envelope Eave Height | |
|---|---|---|---|---|---|---|
| | | | 3.7 m | 12.2 m | 3.7 m | 12.2 m |
| **Low Variability in MWFRS Capacity** | | | | | | |
| Normal to Ridge (90°) | Normal to Ridge | Total | 4.97 | 4.42 | 2.47 | 3.25 |
| | | Left R. | 4.73 | 4.25 | 2.42 | 3.26 |
| | | Right R. | 4.65 | 4.15 | 2.35 | 3.17 |
| Oblique (45°) | Normal to Ridge | Total | 4.68 | 4.22 | 2.31 | 3.05 |
| | | Left R. | 5.76 | 5.14 | 3.35 | 3.85 |
| | | Right R. | 3.50 | 3.20 | 1.35 | 2.22 |
| | Parallel to Ridge | Total | 5.79 | 4.88 | 4.29 | 3.45 |
| | | Left R. | 4.45 | 3.75 | 2.99 | 2.50 |
| | | Right R. | 6.88 | 5.97 | 5.47 | 4.46 |
| Parallel to Ridge (0°) | Parallel to Ridge | Total | 4.53 | 3.88 | 2.98 | 2.44 |
| | | Left R. | 4.29 | 3.67 | 2.88 | 2.43 |
| | | Right R. | 4.30 | 3.75 | 2.89 | 2.50 |
| **High Variability in MWFRS Capacity** | | | | | | |
| Normal to Ridge (90°) | Normal to Ridge | Total | 2.67 | 2.60 | 2.21 | 2.40 |
| | | Left R. | 2.66 | 2.60 | 2.23 | 2.42 |
| | | Right R. | 2.65 | 2.58 | 2.21 | 2.40 |
| Oblique (45°) | Normal to Ridge | Total | 2.68 | 2.57 | 2.22 | 2.36 |
| | | Left R. | 2.86 | 2.74 | 2.50 | 2.54 |
| | | Right R. | 2.48 | 2.39 | 1.93 | 2.17 |
| | Parallel to Ridge | Total | 2.82 | 2.71 | 2.61 | 2.48 |
| | | Left R. | 2.64 | 2.53 | 2.38 | 2.28 |
| | | Right R. | 2.99 | 2.87 | 2.83 | 2.67 |
| Parallel to Ridge (0°) | Parallel to Ridge | Total | 2.66 | 2.54 | 2.39 | 2.24 |
| | | Left R. | 2.65 | 2.52 | 2.39 | 2.26 |
| | | Right R. | 2.66 | 2.53 | 2.40 | 2.28 |

## 4. Discussion

For the types of buildings examined in this study, the design base shear produced by the directional procedure was higher than the design base shear produced by the envelope procedure. For wind normal to ridge and the MWFRS oriented normal to ridge, the base shear produced by the directional procedure was larger compared to the base shear following envelope procedure. Based on the wind tunnel tests data, an overall positive load (pressure acting toward the surface) was developed on the windward roof for the 90° and 45° wind directions. However, the envelope procedure produced a negative (suction) load (pressure acting away from the surface) on the windward roof. Thus, in this respect, the directional procedure better matched the wind tunnel response.

Based on the wind tunnel data, the 45° wind direction produced higher horizontal load normal to ridge than did wind normal to ridge wind. The higher loads were due to higher suction on the leeward surfaces, in particular on the leeward roof surface.

For a MWFRS with a low variation in capacity, the directional procedure produced a reliability index that was 6% larger than a target reliability index equal to 3.0 and the envelope procedure produced a reliability index well below the target. However, for a

MWFRS with high variation in capacity, both the directional and envelope procedures produced a reliability index that was lower than the target.

Additional research is needed to verify the applicability of the findings to other types of buildings. This study examined one building configuration, one roof slope angle, and two eave heights. It is recognized that buildings with different width-to-length (B/L) and height-to-length (H/L) aspect ratios could lead to different conclusions (e.g., [17]), as could buildings with lower-sloped and steeper-sloped roofs, and buildings with hip-shaped roofs, among other aspects. Moreover, this study was limited 0°, 45°, and 90° wind directions. Based on the wind tunnel data, the crucial wind loads occurred in the 45° wind direction. This finding suggests that other wind directions (e.g., 30°, 60°) also need to be examined. Intermediate wind directions could slightly decrease beta for both procedures for higher wind tunnel loads, but the relative performance (design loads) will remain constant. Additionally, this study examined buildings in open terrain, but similar studies are needed to examine the response of buildings in suburban terrain (Exposure B). In this study, the $\beta$ values do not represent a specific type of MWFRS. Future research is needed to determine the impact of the type of MWFRS on the reliability of the wind design procedure.

### 5. Conclusions

The MWFRS design wind loads for a set of low-rise gabled roof buildings in open terrain (Exposure C) were calculated using the directional and envelope procedures described in ASCE 7–16. Unlike previous studies, this study focused on the effect of variability in MWFRS capacity on the reliability of the procedures. The design base shear in each principal direction was compared with the calculated base shear from wind tunnel test data, and the reliability index, $\beta$ conditional on the occurrence of the design wind speed, was determined based on a demand-to-capacity ratio equal to 1.0 and a range of variability in the MWFRS capacity. The results indicate that the directional procedure produced a larger design base shear compared to the envelope procedure, primarily due to the difference in external pressure coefficients. The directional procedure also provided a higher $\beta$ compared to the envelope procedure. The directional procedure provided a $\beta$ that met the standard target $\beta$ equal to 3.0 for main wind force resisting systems with low variability in capacity. Neither the directional procedure nor the envelope procedure met the standard target $\beta$ for main wind force resisting systems with high variability in capacity. Although prior studies establish that both MWFRS design procedures produce wind loads that do not necessarily match loads based on wind tunnel data, this study shows that the only directional procedure provided adequate reliability for low-rise buildings with a MWFRS that exhibits well-defined behavior.

**Author Contributions:** Conceptualization, S.M.A.H. and J.P.J.; methodology, S.M.A.H. and J.P.J.; formal analysis, S.M.A.H.; writing—original draft preparation, S.M.A.H. and J.P.J.; writing—review and editing, S.M.A.H. and J.P.J.; supervision, J.P.J.; funding acquisition, J.P.J. All authors have read and agreed to the published version of the manuscript.

**Funding:** This research was funded in part by the Insurance Institute for Business and Home Safety, and the University of Wyoming.

**Data Availability Statement:** The data, models, and code generated or used during the study are available from the corresponding author by reasonable request.

**Conflicts of Interest:** The authors declare no conflict of interest. The funders had no role in the design of the study; in the collection, analyses, or interpretation of data; in the writing of the manuscript, or in the decision to publish the results.

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
