# Peer review of "Comparison of Wind Tunnel Test Data for Low-Rise Buildings with Main Wind Force Resisting System Design Procedures"

_buildings, doi:10.3390/buildings11080342_

Round 1

Reviewer 1 Report

The paper is well-organized and written. I have few suggestions to improve the readability and the impact of the work as follows:

  1. I suggest updating the literature review in the introduction section. This will help to highlight the novelty of the work. I also suggest not using mass citation in the review (e.g. line 74-75).
  2. The quality of the figures is very good, except for Figure 6. I suggest removing the borderline and increasing the resolution.
  3. The last paragraph of the Conclusion section is presenting the limitations of the work. I suggest moving it to the previous section as a part of the sub-section of the Discussion Section. 
  4. The application of the results needs to be further elaborated in the Conclusion section. What is the contribution of the work to the field?

Author Response

We thank the reviewer for their review and excellent comments. Please find below a point-by-point response to their comments.

  1. We have revised the literature review in the introduction section to highlight the primary novelty of the work (an evaluation of the directional and envelope procedures that accounts for variability in MWFRS capacity) compared to prior work (which has focused primarily on the spatial distribution). We also eliminated the mass citations as much as possible.
  2. We removed the boarder.
  3. Thank you for the suggestion we moved the paragraph to the end of the discussion section. 
  4. Great suggestion. We have revised the conclusions. The main contribution of this work is an understanding of the adequacy of the current (ASCE 7-16) MWFRS design procedures for common low-rise structures including the effect of variability in MWFRS capacity on the reliability.

Reviewer 2 Report

The paper is well written and the context is important for structural design practice. 

Re-write the equations and special characters so they are written as text, not as pictures.

The resolution of Figures 4, 5, and 6 is not adequate, please increase their resolution. 

A suggestion - Tables could be placed closer to the main text which is describing them.

Author Response

We thank the reviewer for their review and excellent comments. Please find below a point-by-point response to their comments. We have revised the paper accordingly.

We changed the manuscript format to address the comments on formatting:

  • The equations are now written and text instead of images.
  • The resolution of Figures 4, 5, and 6 has been increased.
  • We have placed tables as close as possible to the main text in which they are first cited. Please recognize that we were required to upload a version with track-changes and this requirement may affect the placement of tables and figures. Thank you for your understanding.

Reviewer 3 Report

In this study, both the directional and envelope procedures which are described in ASCE 7-16 were employed to calculate the MWFRS design wind loads for one building configuration. The paper is well written and the topic is interesting. However, there are several points that need to be addressed.

Major comments:

  • The reviewer is wondering about the significant contributions of this study. In fact, several other studies have already demonstrated that in general the envelope procedure produces lower design wind pressures, design base shear and reliability index compared to the directional procedure. So the authors need to explicitly identify the major differences of this study compared to other similar research papers.
  • As it was highlighted in the conclusion section, the proposed study is only applicable to one building configuration, and few wind directions along with one type of terrain. If the wind directionality was at least thoroughly investigated, it would make the paper more robust and differentiate it from other similar papers to investigate if the obtained results are holding with the change in this factor.

Minor comments:

  • Please check the paper carefully for typos (e.g., the sentence in Lines 95-96)
  • It seems that there is an error in Fig. 1a). The width is 24.4 m and not 12.2 m.
  • Some mathematical symbols are not clear in the paper (e.g., line 164)

Author Response

We thank the reviewer for their review and excellent comments. Please find below a point-by-point response to their comments.

Response to major comments:

  • We have revised the introduction and conclusions to clearly state the contributions of this study relative to prior studies. We recognized that other studies demonstrate that the envelope procedure generally produces lower design loads compared to the directional procedure, but prior studies were primarily concerned with the spatial distribution of wind loads. No prior study examined how the variability in MWFRS capacity affects their reliability index. We have revised the paper to specifically identify this in the both the introduction and in the conclusions.
  • We agree with the reviewer that a study of directionality would be appropriate. We limited the study to 0, 45, and 90 deg. wind directions because the focus of the paper is on the relative performance of the envelope and directional procedures. Intermediate wind directions could slightly decrease beta for both procedures for higher wind tunnel loads, but the relative performance (design loads) will remain constant.

Response to minor comments:

  • Thank you for pointing this out. We proof read the paper again to eliminate typos.
  • Thank you for catching this. It is has been corrected.
  • We have revised the format of the paper so that mathematical symbols are clearly discernable by inserting each symbol as text instead of a graphic. 

Reviewer 4 Report

It looks good on this manuscript. I believe that the current version of the manuscript is suitable for publication in Buildings.

Author Response

We thank the reviewer for their positive review of our paper.

Reviewer 5 Report

ID: buildings-1229255

Title: Comparison of wind tunnel test data for low-rise buildings with main wind force resisting system design procedures

This paper has pointed out that there exists a gap in deterring MWFRS design wind loads for low-rise buildings during the design of the MWRFS process. Then, they evaluate the directional and envelope procedures for a set of low-rise enclosed buildings.

This paper is quite interesting owing to investigate MWFRS design wind loads base shear force for a set of low-rise buildings using the directional and envelope procedures of ASCE 7-16 using wind tunnel test data. Then they have found that the base shear force of the directional procedure is larger than that of the envelop procedure due to the difference in external pressure coefficients.

Minors:

  1. Line 20: “w” of wind loads should not be bold.
  2. Line 82 : In Table 1, If you provide image or drawings for two example buildings, it will be helpful.
  3. Line 188: Figure 4 is not clear. Please provide a clear figure.
  4. Line 313: Figure 6 is not clear. Please provide a clear figure.

Author Response

We thank the reviewer for their review and excellent comments.

We have addressed their comments as follows:

  • The “w” of wind loads has been fixed (it is no longer in bold font).
  • Drawings of the two example buildings have been added to Table 1. Thank you for the great suggestion.
  • We have replaced the previous images in the figures with higher-resolution images.

Round 2

Reviewer 3 Report

The revised paper can be can be accepted in present form

Author Response

Thank you for reviewing the revised paper.